

# Two new species of Erythroneurini (Hemiptera, Cicadellidae, Typhlocybinae) from southern China based on morphology and complete mitogenomes

Ni Zhang[1,2], Jinqiu Wang[1,2], Tianyi Pu[1,2], Can Li[3] and Yuehua Song[1,2]

[1] School of Karst Science, Guizhou Normal University, Guiyang, China
[2] State Engineering Technology Institute for Karst Desertification Control, Guizhou Normal University, Guiyang, China
[3] Guizhou Provincial Key Laboratory for Rare Animal and Economic Insect of the Mountainous Region/Guizhou Provincial Engineering Research Center for Biological Resources Protection and Efficient Utilization of the Mountainous Region, Guiyang University, Guiyang, China

Corresponding author
Yuehua Song, songyuehua@163.com

## ABSTRACT

Erythroneurine leafhoppers (Hemiptera, Cicadellidae, Typhlocybinae, Erythroneurini) are utilized to resolve the relationship between the four erythroneurine leafhopper (Hemiptera, Cicadellidae, Typhlocybinae, Erythroneurini): *Arboridia* (*Arboridia*) *rongchangensis* sp. nov., *Thaia* (*Thaia*) *jiulongensis* sp. nov., *Mitjaevia bifurcata Luo, Song & Song, 2021* and *Mitjaevia diana Luo, Song & Song, 2021*, the two new species are described and illustrated. The mitochondrial gene sequences of these four species were determined to update the mitochondrial genome database of Erythroneurini. The mitochondrial genomes of four species shared high parallelism in nucleotide composition, base composition and gene order, comprising 13 protein-coding genes (PCGs), 22 transfer RNAs (tRNAs), two ribosomal RNAs (rRNAs) and an AT control region, which was consistent with majority of species in Cicadellidae; all genes revealed common trait of a positive AT skew and negative GC skew. The mitogenomes of four species were ultra-conservative in structure, and which isanalogous to that of others in size and A + T content. Phylogenetic trees based on the mitogenome data of these species and another 24 species were built employing the maximum likelihood and Bayesian inference methods. The results indicated that the four species belong to the tribe Erythroneurini, *M. diana* is the sister-group relationship of *M. protuberanta* + *M. bifurcata*. The two species *Arboridia* (*Arboridia*) *rongchangensis* sp. nov. and *Thaia* (*Thaia*) *jiulongensis* sp. nov. also have a relatively close genetic relationship with the genus *Mitjaevia*.

## INTRODUCTION

The tribe Erythroneurini (*Young, 1952*) is the largest tribe leafhopper subfamily Typhlocybinae, (Hemiptera, Cicadellidae) comprising 209 genera and 2,027 described species (*Dmitriev et al., 2022*) and is widely distributed in all major zoogeographic regions of the world (*Chen et al., 2021*). As plant sap sucking insects they can damage fruit trees

and vegetables, and their small size makes them difficult to detect and identify (*Ghauri, 1967*; *Ghauri, 1974*; *Ghauri, 1975*; *Knight, 1974*). Damage to plants is by egg laying and as virus vectors of plant pathogens (*Womack & Schuster, 1986*; *Bellota, 2011*; *Bosco et al., 2007*). Moreover, erythroneurine species have adopted to various habitats and plants such as trees, rocks, grasslands, sandy substrates, and bushy areas, *etc* (*Morris, 1971*; *Roddee, Kobori & Hanboonsong, 2018*). The species of Erythroneurini have been divided and classified by researchers in different ways, as a result of high morphological diversity and wide geographical distributions.

China is the country with the most widely distributed, fully developed and most complete types of karst landforms in the world, which are mainly concentrated in carbonate outcropping areas, of which Guangxi, Guizhou, and eastern Yunnan account for the largest area (*Xiong et al., 2008*). Guizhou is an important part of the Yunnan-Guizhou Plateau and is believed to be the most well-developed representative of karst areas. The terrain is violently undulating, the types of landforms are diverse, and the composition of surface material and soil types is complex. Additionally, the climate of this area is warm and humid, with small annual temperature changes, warm in winter and cool in summer (*Zhu et al., 2020*), resulting in high biodiversity (plants, insects, birds, snails and bats), except that singular and goodliness natural landscape (*Luo et al., 2016*; *Zhu, 2007*). Many new species of Erythroneurini were discovered in karst regions from Guizhou (*Chen et al., 2020*; *Song, Li & Xiong, 2011*; *Zhang, Song & Song, 2021*).

Phylogenetic relationships of major lineages of Cicadellidae have been researched for many of years (*Skinner et al., 2019*; *Wang et al., 2020a*; *Wang et al., 2020b*; *Lu et al., 2021*; *Yan et al., 2022*; *Hu et al., 2023*; *Cao et al., 2023*). More recently, diverse markers have been applied to perform phylogenetic inferences of Hemiptera, which consist of shape characteristic, mitochondrial genes, nuclear genes and a combination of them, together with transcriptomes on the basis of next-generation sequencing (*Almeida et al., 2009*; *Wang et al., 2010*; *Yao et al., 2021*). With the purpose of confirming the results of traditional classification of Eurythroneurini we also use molecular markers. Based on morphological and molecular data, Erythroneurini has been divided into 209 genera (*Dmitriev et al., 2022*). However, in most instances, short time intervals between speciation events generated incongruous divergence in morphological features and molecular markers (*Chen, Li & Song, 2021*).

The relationships among the multiple species of Cicadellidae were established by means of morphological characters, and a few nuclear genes and mitochondrial sequences (*Longo et al., 2017*; *Jiang et al., 2021*). However, despite the tendency to expand genome coverage, the number of specimens that can be collected is relatively limited, and existing species were chosen to conduct genetic sequencing, as it is impossible to establish a relatively complete molecular identification of the family. Therefore, in our work, the mitochondrial genomes of two new species and two known species (*Mitjaevia bifurcata*, *Mitjaevia diana*) were picked *via* Sequencing Technology to provide a comprehensive comparative analysis of mitochondrial gene structure. We propose a hypothesis that phylogenetic trees based on mitochondrial genomes can better validate the accuracy of traditional classification. Phylogenetic trees based on the mitochondrial genomes of *A. (A.) rongchangensis* sp. nov.,
**Table 1** Study sites and dates for leafhopper sample collection in this study.

| Species | Locality | Collector | Latitude | Longitude | Date |
|---|---|---|---|---|---|
| A. (Arboridia) rongchangensis sp. nov. | Rongchang, Chongqing | Guimei Luo | 29°25′43″N | 105°39′21″E | 14 Aug 2021 |
| T. (Thaia) jiulongensis sp. nov. | Jiulongpo, Chongqing | Weiwen Tan | 29°28′36″N | 106°25′11″E | 14 Aug 2021 |
| M. bifurcata | Bijie, Guizhou | Zhouwei Yuan | 27°14′51″N | 105°5′52″E | 27 May 2019 |
| M. diana | Huajiang, Guizhou | Zhouwei Yuan | 25°41′36″N | 105°37′46″E | 29 May 2019 |

*T.* (*T.*) *jiulongensis* sp. nov., *M. bifurcata* and *M. diana* and another 24 species were built adopting the Bayesian inference and maximum likelihood methods. This research will enrich the mitochondrial gene bases of the erythroneurine leafhoppers and improve the accuracy of the traditional classification.

## MATERIAL AND METHODS

### Leafhopper collections and species identification based on the morphology

The species of leafhopper are collected according to Table 1. The specimens were preserved in absolute ethanol. Images of the appearance and genitalia of species were taken by a KEYENCE VHX-5000 digital microscope. Male/female specimens were identified under a stereoscope, and the whole abdomen of the specimens was separated and moistened in a hot 10% NaOH solution. Afterward, the abdomen was washed with ordinary water, blotted up with qualitative filter paper, and transferred to a clean glass slide with a drop of glycerin. Genital dissections were dissected in glycerin to inhibit parts from drying out. Then, they were viewed and plotted by way of Olympus SZX16 and BX53 microscopes. The remaining specimen was stored in 95% ethanol and put in a refrigerator at −20 °C. The analyzed specimens were examined using Olympus SZX16 dissecting microscope and Olympus BX53 stereoscopic microscopes respectively and identified by Prof. Yuehua Song. All specimens inspected are reserved in the School of Karst Science, Guizhou Normal University, China (GZNU).

### DNA extraction, mitogenome sequencing and assembly

Extraction of DNA originated from the whole body removing the abdomen and wings. The bodies were incubated at 56 °C for 6 h for complete lysis and total genomic DNA was eluted in 50 µL double-distilled water (ddH$_2$O), and the remaining other steps were performed according to the manufacturer's protocol. Genomic DNA was stored at −20 °C. The whole mitochondrial genomes of *A.* (*A.*) *rongchangensis* sp. nov. and *T.* (*T.*) *jiulongensis* sp. nov. were sequenced at Berry Genomics (Beijing, China) by an Illumina Novaseq 6000 platform (Illumina, Alameda, CA, USA) using 150 bp paired-end reads. Firstly, the obtained sequence reads were filtered following *Zhou et al. (2013)*, the remaining high-quality reads were assembled by an iterative De Bruijin graph *de novo* assembler, the IDBA-UD toolkit, with a similarity threshold of 98%, and k values of 40 and 160 bp (*Peng et al., 2012*). The mitogenome was initially assembled by Geneious Prime v 2021.1.1, and then manually proofread based on sequencing peak figures.

The complete mitochondrial genomes of *M. Bifurcata* and *M. diana* were sequenced at Bio-Transduction Lab Co.Ltd. (Wuhan, China) by Sanger sequencing. PCR primers were designed according to conserved region sequences and used to amplify the mitochondrial DNA sequence in PCR reactions (Tables 2 and 3). The PCR reaction was performed using the LA Taq polymerase. The thermal cycling conditions comprised an initial denaturation step at 94 °C for 2 min, then 35 cycles of denaturation at 94 °C for 30 s, 30 s for annealing at 55 °C, and elongation at 72 °C for 1 min/kb, followed by the final extension at 72 °C for 10 min. The PCR products were purified and sequenced using an ABI 3730 automatic sequencer. After quality-proofing of the obtained DNA fragments, and BLASTed were used to confirm that the amplification is the actual target sequence (*Meng et al., 2013*; *Yu, Wu & Han, 2017*). The complete mitogenome sequence was assembled manually through DNAStar v7.1 (*Burland, 2000*).

## Genome annotation and analyses

First of all, raw mitogenomic sequences were entered into MITOS web servers (http://mitos.bioinf.uni-leipzig.de/index.py, accessed on 15 Jun 2021) in an effort to fix the rough boundaries of genes. Accurate locations of protein-coding genes (PCGs) were determined by seeking ORFs (employing genetic code 5, the invertebrate mitochondrion). All tRNAs were characeried by using tRNAscan SE v. 1.21 and ARWEN (*Lowe & Eddy, 1997*; *Laslett & Canbck, 2008*). The precise boundaries of *rrnL* and *rrnS* were defined by homologous comparison. Genomes manually annotated were parsed and extracted by means of PhyloSuite, and GenBank (NCBI) submission files and organization tables for mitogenomes were also created through the same software (*Zhang et al., 2020*).

The mitogenomic circular map was generated by OrganellarGenomeDRAW (OGDRAW) version 1.3.1 (https://chlorobox.mpimp-golm.mpg.de/OGDraw.html, accessed on 3 March 2023) (*Greiner, Lehwark & Bock, 2019*). Intergenic spacers and overlapping regions between genes were performed manually. The nucleotide base composition, codon usage, as well as values of A + T content were calculated with MEGA 11.0 (*Tamura, Stecher & Kumar, 2021*). The bias of nucleotide composition was computed according to AT skew = [A − T]/[A + T] and GC skew = [G − C]/[G + C] (*Perna & Kocher, 1995*). Additionally, the nucleotide diversity (Pi) and nonsynonymous (Ka)/synonymous (Ks) mutation rate ratios were operated by DNAsp 6.0 (*Rozas et al., 2017*).

## Phylogenetic analysis

A molecular phylogenetic analysis was constructed on the basis of mitogenomes of 28 species and two species regarded as outgroups (Table 4). All complete mitochondrial sequences were selected to accomplish phylogenetic analyses. The Gblocks version 0.91b was adopted to clean out the gaps and fuzzy-alignment sites, and all alignments were verified and revised in MEGA 11.0 prior to phylogenetic analysis (*Tamura, Stecher & Kumar, 2021*). The phylogenetic trees were constructed by introducing two methods both the maximum likelihood (ML) method and the Bayesian Inference (BI) method (*Nguyen et al., 2015*; *Zhou et al., 2011*). The ML analysis was performed with IQ-TREE under a ML

**Table 2  Primers used for amplification of the mitochondrial genome of *M. bifurcata*.**

| Fragment no. | Gene or region | Sequence (5′–3′) | Length (bp) |
|---|---|---|---|
| F1 | tRNA-Met-COX1 | GCTAACTTAAGCTATTAGGTTC CGTATGTTAATTACTGTTGTG | 1,720 |
| F2 | COX1 | CTGGTTGAACAGTTTACCC CATCTAAAAACCTTAATACC | 598 |
| F3 | COX1-ATP6 | GAGTCATTTGGTTATATTGG GAAATTTCTCCTTGAAGAGA | 1,949 |
| F4 | ATP6 | CAGTTTTTGATCCTTGTACTG GCCTGCAATTATGTTAGCAG | 473 |
| F5 | ATP6-COX3 | GACATTTAGTACCTGTTGGTACG CTCAAATCCTACATGATGCC | 1,206 |
| F6 | COX3-ND5 | CAGGTGTTTCTATTACATGAG CGTTTAGGGGATATTGGTCTG | 2,428 |
| F7 | ND5 | TGCAGTTACCAGGGTTGAAG GTTAGGTTGAGATGGCTTGG | 328 |
| F8 | ND5-ND4 | CCAATATCCCCTAAACGGTTAG GTTTACTACAAGGAGATGTA | 1,067 |
| F9 | ND4 | CTGAAGAACATAACCCATGAG GATTACCAAAAGCGCATGTTC | 330 |
| F10 | ND4-12S | GTGAATACCAAACATAACTG AAGCAGACATGTGTTACT | 5,295 |
| F11 | 12S | CCAGTACAATTACTTTGTTACG CTTTAACATTAATAGTTTATTTTC | 385 |
| F12 | 12S-ND2 | CAATTAAGATACAGGTTCCC GAGTGCAAAAGAGGCAGGAATG | 3,235 |

+ rapid bootstrap (BS) algorithm with 10,000 replicates used to calculate bootstrap scores for each node (BP). The BI analysis was carried out using MrBayes 3.2.7 elected GTR + G + I as the optimal model, running 10 million generations, sampling every 1000 trees, 25% of samples were abandoned as burn-in.

## Nomenclature

The electronic version of this article in Portable Document Format (PDF) will represent a published work according to the International Commission on Zoological Nomenclature (ICZN), and hence the new names contained in the electronic version are effectively published under that Code from the electronic edition alone. This published work and the nomenclatural acts it contains have been registered in ZooBank, the online registration system for the ICZN. The ZooBank LSIDs (Life Science Identifiers) can be resolved and the associated information viewed through any standard web browser by appending the LSID to the prefix http://zoobank.org/. The LSID for this publication is: http://zoobank.org/urn:lsid: zoobank.org:pub:2E7A89DA-21F4-41D1-B9DF-A33BCD5F3086; http://zoobank.org/urn: lsid:zoobank.org:pub:7710E194-8D88-413B-B3D2-7AF0B0777BE7. The online version of

**Table 3  Primers used for amplification of the mitochondrial genome of *M. diana*.**

| Fragment no. | Gene or region | Primer name | Sequence (5′–3′) | Length (bp) |
|---|---|---|---|---|
| F1 | tRNA-Met-COX1 | D2F1 | GCTAATTTAAGCTATTAGGTTC | 2,197 |
| | | D2R1 | GTGACTCCATGTATTGTAGC | |
| F2 | COX1-ATP6 | D2F2 | GGTTTGTTGTTTGGGCTCATC | 1,927 |
| | | D2R2 | AGTTGGATACCCCTGTAAGG | |
| F3 | ATP6 | D2F3 | TGTTTTCAGTATTTGACCCTTG | 481 |
| | | D2R3 | TGCCCTGCAATTATATTAGC | |
| F4 | ATP6-COX3 | D2F4 | GACATTTAGTTCCGGTAGGA | 957 |
| | | D2R4 | GAAGGTTATACATTCGAATCC | |
| F5 | COX3 | D2F5 | TAGCAACAGGATTTCATGGA | 142 |
| | | D2R5 | TCTACAAAGTGTCAATACCAAG | |
| F6 | COX3-ND5 | D2F6 | TAGTATCTGGGATTCGAATG | 2,159 |
| | | D2R6 | ATGTCTTTTGGTAGTTGAC | |
| F7 | ND5 | D2F7 | GGAAGAATGAACTAGAGATG | 314 |
| | | D2R7 | TGCTGGGTTGAGATGGTTTAG | |
| F8 | ND5-ND4 | D2F8 | CCTAAACGATTAGTTAAGCAAG | 1,103 |
| | | D2R8 | GGTATTCATTAAACTTAGTAGG | |
| F9 | ND4 | D2F9 | CCAGATGAACATAAACCGTGAG | 326 |
| | | D2R9 | CCAAAAGCTCATGTTCAAGC | |
| F10 | ND4-CYTB | D2F10 | CAAAGATACTTATAACTCGG | 2,222 |
| | | D2R10 | CTGTGATGTGTAGAAAGAAG | |
| F11 | CYTB | D2F11 | GTAATCACTAATTTACTATCTGC | 383 |
| | | D2R11 | CATTCTGGTTGAATATGAATC | |
| F12 | CYTB-16S | D2F12 | GATTTACTGGGAATTGTAATTAC | 1,773 |
| | | D2R12 | GTTACCTTAGGGATAACAGC | |
| F13 | 16S | D1F13 | CACCGATTTGAACTCAAATC | 987 |
| | | D1R13 | GGTTTTGTACCTTTTGTATTAGG | |
| F14 | 16S-12S | D1F14 | GTAAAGATTATCCCTTAC | 639 |
| | | D1R14 | GTTAGGTCAAGGTGCAGT | |
| F15 | 12S | D1F15 | CTTTGTTACGACTTATCTC | 419 |
| | | D1R15 | TTAGGATTAGATACCCTAT | |
| F16 | 12S-ND2 | D1F16 | GTGGTTTATCAATTAAGAAAC | 2,976 |
| | | D1R16 | GCTTAATTCCAAGCCACACC | |

this work is archived and available from the following digital repositories: PeerJ, PubMed Central SCIE and CLOCKSS.

**Table 4** List of mitochondrial genomes analyzed in the present.

| Subfamily/Tribe | Species | Length (bp) | Accession number | Reference |
|---|---|---|---|---|
| Typhlocybinae/ Typhlocybini | *Eurhadina acapitata* | 15,419 | MZ457331.1 | Direct submission |
| | *Eurhadina jarrayi* | 15,332 | MZ014455.1 | *Lin, Huang & Zhang (2021)* |
| | *Eurhadina dongwolensis* | 15,708 | MZ457332.1 | Direct submission |
| | *Eurhadina fusca* | 15,302 | MZ983367.1 | Direct submission |
| | *Agnesiella kamala* | 15,209 | MZ457327.1 | Direct submission |
| | *Agnesiella roxana* | 15,901 | MZ457328.1 | Direct submission |
| | *Eupteryx adspersa* | 15,178 | MZ014454.1 | *Lin, Huang & Zhang (2021)* |
| | *Eupteryx minuscula* | 16,944 | MN910279.1 | *Yang et al. (2020)* |
| | *Eupteryx gracilirama* | 17,173 | MT594485.1 | *Yuan et al. (2021a)* and *Yuan et al. (2021b)* |
| Typhlocybinae/ Erythroneurini | *Limassolla emmrichi* | 14,677 | MW272458.1 | *Yan et al. (2022)* |
| | *Limassolla lingchuanensis* | 15,716 | NC_046037.1 | *Yuan, Li & Song (2020)* |
| | *Limassolla* sp. | 17,053 | MT683892.1 | *Zhou, Dietrich & Huang (2020)* |
| | *Mitjaevia bifurcata* | 16,589 | OK448488.1 | Direct submission |
| | *Mitjaevia protuberanta* | 15,472 | NC_047465.1 | *Yuan, Li & Song (2020)* |
| | *Mitjaevia diana* | 16,183 | OK448489.1 | Direct submission |
| | *Mitjaevia dworakowskae* | 16,399 | MT981880.1 | *Chen, Li & Song (2021)* |
| | *Mitjaevia shibingensis* | 15,788 | MT981879.1 | *Chen, Li & Song (2021)* |
| | *Arboridia* (*Arboridia*) *rongchangensis* sp. nov. | 15,596 | OQ404948.1 | Direct submission |
| | *Thaia* (*Thaia*) *jiulongensis* sp. nov. | 15,676 | OQ630475.1 | Direct submission |
| | *Elbelus tripunctatus* | 15,308 | MZ014452.1 | *Lin, Huang & Zhang (2021)* |
| | *Empoascanara sipra* | 14,827 | NC_048516.1 | *Tan et al. (2020)* |
| | *Empoascanara wengangensis* | 14,830 | MT445764.1 | *Chen et al. (2021)* |
| | *Empoascanara dwalata* | 15,271 | MT350235.1 | *Chen et al. (2016)* |
| | *Empoascanara gracilis* | 14,627 | MT576649.1 | *Chen et al. (2021)* |
| Typhlocybinae/ Empoascini | *Empoasca onukii* | 15,167 | NC_037210.1 | *Song, Zhang & Zhao (2019)* |
| | *Empoasca vitis* | 15,154 | NC_024838.1 | *Zhou et al. (2016)* |
| | *Empoasca flavescens* | 15,152 | MK211224.1 | *Luo et al. (2019)* |
| | *Empoasca serrata* | 15,131 | MZ014453.1 | *Lin, Huang & Zhang (2021)* |
| | *Bothrogonia ferruginea* | 15,262 | KU167550.1 | *Yu et al. (2019)* |
| | *Iassus dorsalis* | 15,176 | NC_046066.1 | *Wang et al. (2020b)* |

# RESULTS AND DISCUSSION

## Taxonomy based on morphology

*Arboridia* (*Arboridia*) *rongchangensis* Zhang & Song, sp. nov. (Figs. 1–2)

Description.

Dorsum dark brownish (Figs. 1A and 1C). Color pattern brown. Vertex with a pair of dark preapical spots. Face yellowish white, frontoclypeus dark (Figs. 1B and 1D).

Head narrower than pronotum (Figs. 1A and 1C). Crown fore margin weakly produced medially. Face, with anteclypeus narrow and pale, and frontoclypeus dark (Figs. 1B and

1D). Pronotum wide, scutellum with lateral triangles (Figs. 1A and 1C). Forewings without spots or markings.

Male genitalia. Pygofer dorsal appendage simple, without branch, hook-like apically (Figs. 2E and 2F). Subgenital plate with two macrosetae on lateral surface, and row of peg-like setae from subbase to apex, and several microsetae scattered on apical portion (Fig. 2D). Style long and slender, with two points apically; preapical lobe obtuse and distinct (Fig. 2A). Aedeagus with a large lamellate process arising from base of aedeagal shaft ventrally; aedeagal shaft broad and flat, slightly bifurcated at apex; gonopore subapical on ventral surface; preatrium little longer than shaft (Figs. 2B and 2C). Connective V-shaped, with arms long (Fig. 2G).

Male abdominal apodemes small, not exceeding 3rd sternite (Fig. 2H).

Measurement. Male length 3.0∼3.1 mm, female length 3.1∼3.2 (including wings).

Specimen examined. Holotype: ♂, CHINA, Chongqing, Rongchang, 14 VIII 2021, coll. Guimei Luo. Paratypes: 11 ♂♂, 32 ♀♀, same data as holotype.

Remarks. This species is similar to *Arboridia reniformis Song & Li (2013)*, but differs in having the aedeagal shaft distinctly bifurcate apically.

Etymology. The new species is named after its type locality: ''Rongchang'' county, Chongqing, China.

### *Thaia* (*Thaia*) *jiulongensis* Zhang & Song, sp. nov. (Figs. 3–4)

Description. Vertex yellow, light brownish in middle apically (Figs. 3A and 3C). Face yellowish brown (Figs. 3B and 3D). Pronotum orange brown. Scutellum with anterior margin yellow and posterior part milky yellow; lateral triangles brownish black (Figs. 3A and 3C).

Face with anteclypeus ovoid (Figs. 3B and 3D). Pronotum, with pair of large triangular impressions, posterior and anterior part lighter, yellowish (Figs. 3A and 3C).

Male abdominal apodeme small, not surpassing 3rd sternite (Fig. 4I).

Male genitalia. Pygofer lobe with scattered fine microsetae on dorsal surface, with dorso-caudal margin angulated (Fig. 4F). Anal tube with well-developed basal appendages, extending ventro-caudally (Fig. 4G). Subgenital plate broadened at subbase, provided with three macrosetae on lateral surface at midlength, numerous peg-like small setae along dorsal margin from near midlength part to apex; several small setae scattered apically (Figs. 4E and 4F). Style slender apically, preapical lobe well developed (Fig. 4A). Aedeagus expanded at base in ventral view, with pair of long basal process arising from preatrium ventrally, which slim and curved, tapering towards apex; gonopore apical on ventral surface (Figs. 4B, 4C and 4D). Connective V-shaped, without central lobe, lateral arms long and slim (Fig. 4H).

Specimen examined. Holotype: ♂, CHINA, Chongqing, Jiulongpo District, Zhongliang Yunling Forest Park, 14 VIII, 2021. coll. Weiwen Tan. Paratypes: 5 ♂♂, 15 ♀♀, same data as holotype.

Measurement. Body length ♂ 3.0∼3.2 mm; ♀ 3.0∼3.2 mm (including wings).

Remarks. This species is similar to *Thaia* (*Thaia*) *barbata Dworakowska (1979)*, but can be distinguished from the latter species by from the shape of the adedeagus, with a small

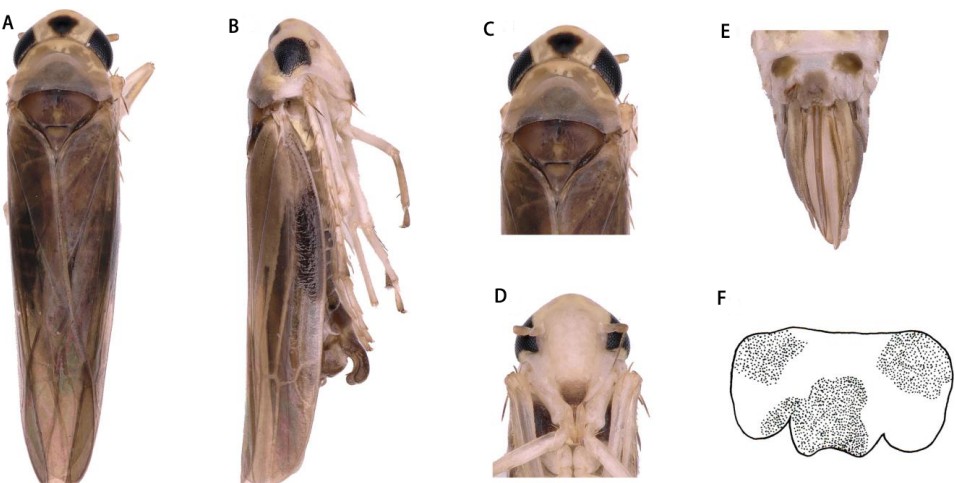

**Figure 1** **Arboridia (Arboridia) rongchangensis Zhang & Song, sp. nov.** (A) Dorsal habitus. (B) Lateral habitus. (C) Head and thorax, dorsal view (D) Face. (E) Terminalia of female, ventral view. (F) Sternite VII of female, ventral view.

subapical protrusion and, the anal tube appendages, which are particularly wide compared to *T.* (*T.*) *barbata* Dworakowska (1979). Connective arms are relatively slender.

Etymology. The new species is named after its type locality, Jiulong, Chongqing.

## Taxonomy based on molecular data
## Organization and composition of the genome

The complete mitogenomes of *A.* (*A.*) *rongchangensis* sp. nov., *T.* (*T.*) *jiulongensis* sp. nov., *M. bifurcata* and *M. diana* are 15,596, 15676, 16,183 and 16,589 bp, respectively. Both species comprise 13 PCGs, 22 tRNA genes, two rRNA genes, and a control region (CR) (Fig. 5). Two strands, the majority strand (H-strand) and the minority strand (L-strand), exist in the mitochondrial genome. The H-strand consists of 23 genes (nine PCGs, 14 tRNAs) and CR, and meanwhile, the L-strand encompasses 14 genes (four PCGs, eight tRNAs, and two rRNAs).

There are 50 bp, 66 bp, and 70 bp intergenic spaces presented in total length of all the intergenic space ranging from one to 10 bp, one to 13 bp, one to nine bp in *A.* (*A.*) *rongchangensis* sp. nov., *M. bifurcata*, *M. diana*. However, 52 bp intergenic space existed in 11 regions from one to 15 bp in *T.* (*T.*) *jiulongensis* sp. nov. It can be observed that ten genes overlapped by 28 bp in *A.* (*A.*) *rongchangensis* sp. nov., eleven genes overlapped by 31 bp in *T.* (*T.*) *jiulongensis* sp. nov., ten genes overlapped by a grand total of 32 bp in *M. bifurcata*, nine genes overlapped by 40 bp *M. diana* (Table 5). The heavy AT nucleotide bias appears in the mitochondrial genomes in *A.* (*A.*) *rongchangensis* sp. nov., *T.* (*T.*) *jiulongensis* sp. nov., *M. bifurcata* and *M. diana*, the A + T contents are 80.7%, 78.0%, 78.4% and 78.5%, respectively (Table 5).

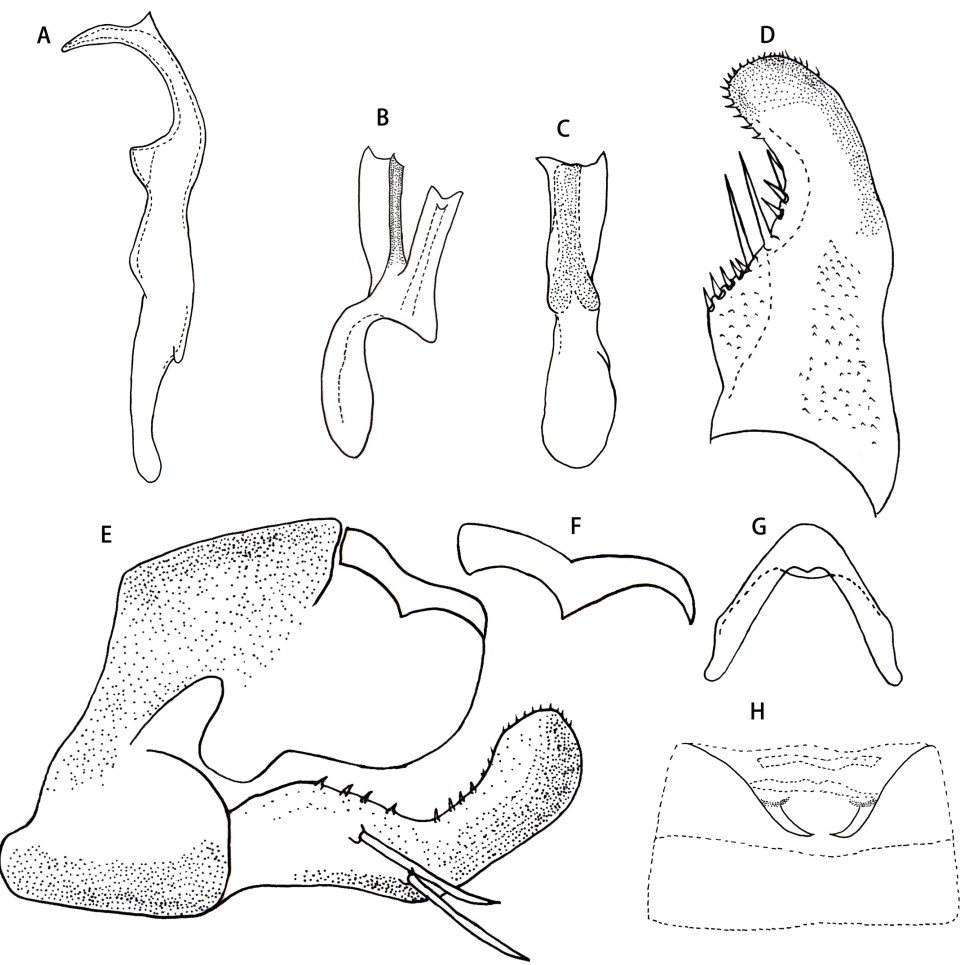

**Figure 2** *Arboridia (Arboridia) rongchangensis* **Zhang & Song, sp. nov.** (A) Style. (B) Aedeagus, lateral view. (C) Aedeagus, ventral view. (D) Subgenital plate. (E) Pygofer lobe, lateral view. (F) Pygofer dorsal appendage, lateral view. (G) Connective. (H) Abdominal apodemes.

## Protein-coding genes and codon usage

As with most other Typhlocybinae, the overall length of 13 PCGs of *A.* (*A.*) *rongchangensis* sp. nov., *T.* (*T.*) *jiulongensis* sp. nov., *M. bifurcata* and *M. diana* are 10,946, 10,968, 10,966 bp and 10,966 bp, 70.3%, 69.8%, 65.20% and 66.1% of the total genome of each species, respectively. In addition, *nad2* and *cox3* in four species have the same start codons and stop codons. The longest PCG is *nad5* (1,675 bp) in *A.* (*A.*) *rongchangensis* sp. nov., the shortest is *atp8* (144 bp) in *M. bifurcata* and *M. diana*. Only four genes (*nad5, nad4, nad4L* and *nad1*) are presented on the J-strand, and the remaining nine genes are presented on the H-strand.

The relative synonymous codon usage (RSCU) values of the 13 PCGs are generalized in Fig. 6. The codon usage analyses of *A.* (*A.*) rongchangensis sp. nov., *T.* (*T.*) *jiulongensis* sp. nov., *M. bifurcata* and *M. diana* revealed that codon UUA-Leu2 (214, 194, 180, 241), AUU-Ile (297, 249, 274, 210), AUA-Met (245, 202, 223, 180) AAU-Asn (273, 239, 236, 256),

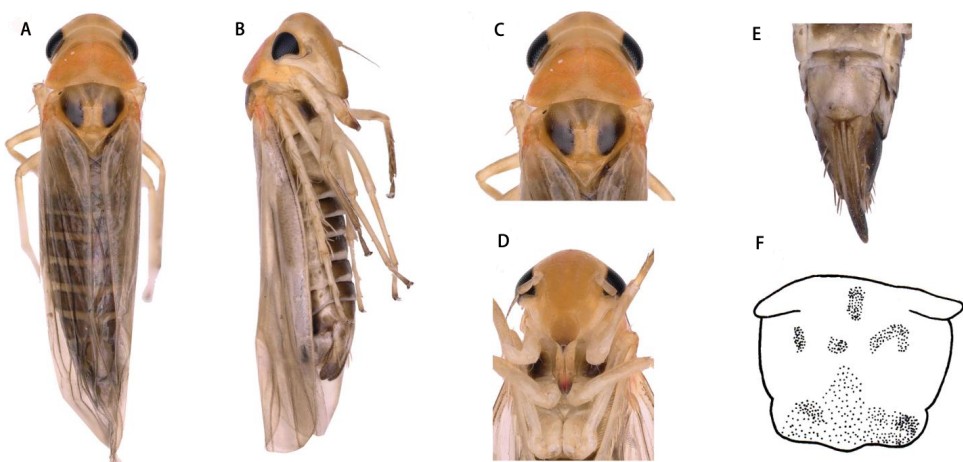

**Figure 3** *Thaia (Thaia) jiulongensis* **Zhang & Song, sp. nov.** (A) Dorsal habitus. (B) Lateral habitus. (C) Head and thorax, dorsal view (D) Face. (E) Terminalia of female, ventral view. (F) Sternite VII of female, ventral view.

and AAA-Lys (290, 280, 227, 238) are the most frequently used. The highest RSCU value of each species occur in UUA-Leu2. The results showed that UUA is the most preferred codon. In addition, it can be seen from the RSCU values of the PCGs that AT is used more frequently than GC.

## Transfer RNA and ribosomal RNA genes

The 22 tRNAs were deconcentrated between two regions, the rRNAs and the protein-coding region. The total tRNA lengths of *A. (A.) rongchangensis* sp. nov., *T. (T.) jiulongensis* sp. nov., *M. bifurcata* and *M. diana* are 1,434, 1,436, 1,441 and 1,455 bp, respectively, which range from 61 to 70 bp in *A. (A.) rongchangensis* sp. nov., 61 to 71 in *T. (T.) jiulongensis* sp. nov., 61 to 71 bp in *M. bifurcata* and 62 to 71 bp in *M. diana* (Table 5). The sequences of most tRNA genes demonstrated the exemplary clover-leaf secondary structure, including four structural domains and a short flexible loop: the acceptor stem, the dihydrouridine stem and loop (DHU), the anticodon stem (DHU) and loop, the thymidine stem and loop (T $\psi$ C), and the variable (V) loop (Fig. S1), as observed in many other leafhoppers mitogenomes. However, the dihydrouridine (DHU) arm of *trnS1* shapes an uncomplicated loop. Additionally, non-Waston-Crick base pairs were harbored the stems of the secondary structures, 21, 27, 21 and 18 weak G-U (or U-G) base pairs are revealed in the tRNAs of *A. (A.) rongchangensis* sp. nov., *T. (T.) jiulongensis* sp. nov., *M. bifurcata* and *M. diana* (Fig. S1). The location of mismatched base pairs in the acceptor arm, DHU arm, T $\psi$ C arm and anticodon arm of tRNA from four species were shown in Table 6. These mismatches could be rectified by way of editing process, and the transport function is not influenced (*Yuan et al., 2021a*; *Yuan et al., 2021b*).

Zhang et al. (2024), *PeerJ*, DOI 10.7717/peerj.16853

**Table 5  Organization of the *A. (A.) rongchangensis* Zhang & Song, sp. nov., *T. (T.) jiulongensis* Zhang & Song, sp. nov., *M. bifurcata* and *M. diana* mitochondrial genome.**

| Gene | Position | | | | Intergenic | | | | Start codon | | | | Stop codon | | | | Strand |
|---|---|---|---|---|---|---|---|---|---|---|---|---|---|---|---|---|---|
| | *A. (Arboridia) rongchangensis* sp. nov. | *T. (Thaia) jiulongensis* sp. nov. | *M. bifurcata* | *M. diana* | | | | | | | | | | | | | |
| trnI | 1–63 | 1–64 | 1–64 | 1–63 | 0 | 0 | 0 | 0 | | | | | | | | | H |
| trnQ | 61–129 | 63–131 | 62–130 | 63–129 | −3 | −2 | −3 | −3 | | | | | | | | | L |
| trnM | 140–208 | 135–203 | 140–207 | 140–207 | 10 | 3 | 9 | 10 | | | | | | | | | H |
| nad2 | 212–1180 | 204–1175 | 208–1179 | 208–1179 | 3 | 0 | 0 | 0 | ATA | ATA | ATA | ATA | TAA | TAA | TAA | TAA | H |
| trnW | 1179–1242 | 1174–1237 | 1178–1240 | 1178–1240 | −2 | −2 | −2 | −2 | | | | | | | | | H |
| trnC | 1235–1300 | 1237–1297 | 1233–1293 | 1233–1294 | −8 | −1 | −8 | −8 | | | | | | | | | L |
| trnY | 1306–1367 | 1298–1372 | 1294–1356 | 1295–1356 | 5 | 0 | 4 | 0 | | | | | | | | | L |
| cox1 | 1369–2907 | 1366–2904 | 1361–2896 | 1361–2896 | 1 | −7 | 0 | 4 | ATT | ATG | ATG | ATG | TAA | TAA | TAA | TAA | H |
| trnL2 | 2911–2977 | 2907–2973 | 2898–2964 | 2898–2964 | 3 | 2 | 1 | 1 | | | | | | | | | H |
| cox2 | 2978–3656 | 2974–3652 | 2965–3643 | 2965–3643 | 0 | 0 | 0 | 0 | ATA | TTG | ATT | ATT | T | T | T | T | H |
| trnK | 3657–3726 | 3653–3723 | 3644–3714 | 3644–3714 | 0 | 0 | 0 | 0 | | | | | | | | | H |
| trnD | 3727–3794 | 3724–3786 | 3719–3782 | 3723–3785 | 0 | 0 | 4 | 8 | | | | | | | | | H |
| atp8 | 3794–3946 | 3786–3938 | 3792–3935 | 3795–3938 | −1 | −1 | 9 | 9 | TTG | TTG | ATG | ATG | TAA | TAA | TAA | TAA | H |
| atp6 | 3943–4593 | 3932–4585 | 3929–4582 | 3932–4585 | −4 | −7 | −7 | −7 | ATA | ATG | ATG | ATG | TAA | TAA | TAA | TAA | H |
| cox3 | 4594–5373 | 4586–5365 | 4583–5362 | 4586–5365 | 0 | 0 | 0 | 0 | ATG | ATG | ATG | ATG | TAA | TAA | TAA | TAA | H |
| trnG | 5380–5441 | 5366–5428 | 5367–5428 | 5371–5432 | 6 | 0 | 4 | 5 | | | | | | | | | H |
| nad3 | 5442–5795 | 5429–5782 | 5429–5782 | 5433–5786 | 0 | 0 | 0 | 0 | ATT | ATT | ATA | ATA | TAA | TAA | TAA | TAA | H |
| trnA | 5797–5857 | 5797–5859 | 5796–5856 | 5792–5853 | 1 | 14 | 13 | 5 | | | | | | | | | H |
| trnR | 5857–5919 | 5864–5924 | 5856–5916 | 5853–5917 | −1 | 4 | −1 | −1 | | | | | | | | | H |
| trnN | 5919–5983 | 5924–5990 | 5916–5979 | 5917–5982 | −1 | −1 | −1 | −1 | | | | | | | | | H |
| trnS1 | 5983–6049 | 5990–6056 | 5979–6046 | 5982–6049 | −1 | −1 | −1 | −1 | | | | | | | | | H |
| trnE | 6052–6118 | 6060–6123 | 6054–6116 | 6059–6122 | 2 | 3 | 7 | 9 | | | | | | | | | H |
| trnF | 6120–6183 | 6139–6203 | 6123–6190 | 6128–6195 | 1 | 15 | 6 | 5 | | | | | | | | | L |
| nad5 | 6184–7858 | 6203–7876 | 6193–7866 | 6197–7870 | 0 | −1 | 2 | 1 | ATT | TTG | TTG | TTG | T | TAA | TAA | TAA | L |
| trnH | 7856–7917 | 7877–7945 | 7867–7931 | 7871–7935 | −3 | 0 | 0 | 0 | | | | | | | | | L |
| nad4 | 7918–9241 | 7949–9248 | 7931–9259 | 7935–9263 | 0 | 3 | −1 | −1 | ATA | ATG | ATG | ATG | T | T | TAA | TAA | L |
| nad4L | 9238–9516 | 9242–9517 | 9253–9531 | 9257–9535 | −4 | −7 | −6 | −6 | ATG | ATT | ATG | ATG | TAG | TAA | TAA | TAA | L |
| trnT | 9519–9584 | 9520–9583 | 9534–9597 | 9538–9601 | 2 | 2 | 2 | 2 | | | | | | | | | H |
| trnP | 9585–9649 | 9584–9651 | 9598–9661 | 9602–9665 | 0 | 0 | 0 | 0 | | | | | | | | | L |
| nad6 | 9652–10137 | 9654–10139 | 9664–10149 | 9668–10153 | 2 | 2 | 2 | 2 | ATG | ATT | ATT | ATT | TAA | TAA | TAA | TAA | H |

**Table 5** (*continued*)

| Gene | Position | | | | Intergenic | | | | Start codon | | | | Stop codon | | | | Strand |
|---|---|---|---|---|---|---|---|---|---|---|---|---|---|---|---|---|---|
| | *A.* (*Arboridia*) *rongchangensis* sp. nov. | *T.* (*Thaia*) *jiulongensis* sp. nov. | *M. bifurcata* | *M. diana* | | | | | | | | | | | | | |
| *cytb* | 10138–11274 | 10142–11278 | 10153–11289 | 10161–11297 | 0 | 2 | 3 | 7 | ATG | ATG | ATG | ATG | TAA | TAG | TAG | TAA | H |
| *trnS2* | 11279–11342 | 11278–11341 | 11288–11343 | 11300–11365 | 4 | −1 | −2 | 2 | | | | | | | | | H |
| *nad1* | 11333–12274 | 11344–12274 | 11344–12285 | 11356–12297 | 10 | 2 | 0 | −10 | ATT | ATT | ATT | ATT | TAA | T | TAA | TAA | L |
| *trnL1* | 12275–12339 | 12275–12342 | 12286–12352 | 12298–12362 | 0 | 0 | 0 | 0 | | | | | | | | | L |
| *rrnL* | 12340–13586 | 12343–13537 | 12353–13538 | 12363–13548 | 0 | 0 | 0 | 0 | | | | | | | | | L |
| *trnV* | 13522–13586 | 13538–13599 | 13539–13605 | 13549–13618 | 0 | 0 | 0 | 0 | | | | | | | | | L |
| *rrnS* | 13587–14320 | 13600–14331 | 13606–14341 | 13619–14356 | 0 | 0 | 0 | 0 | | | | | | | | | L |
| D-loop | 14321–15596 | 14332–15676 | 14342–16813 | 14357–16589 | | | | | | | | | | | | | |

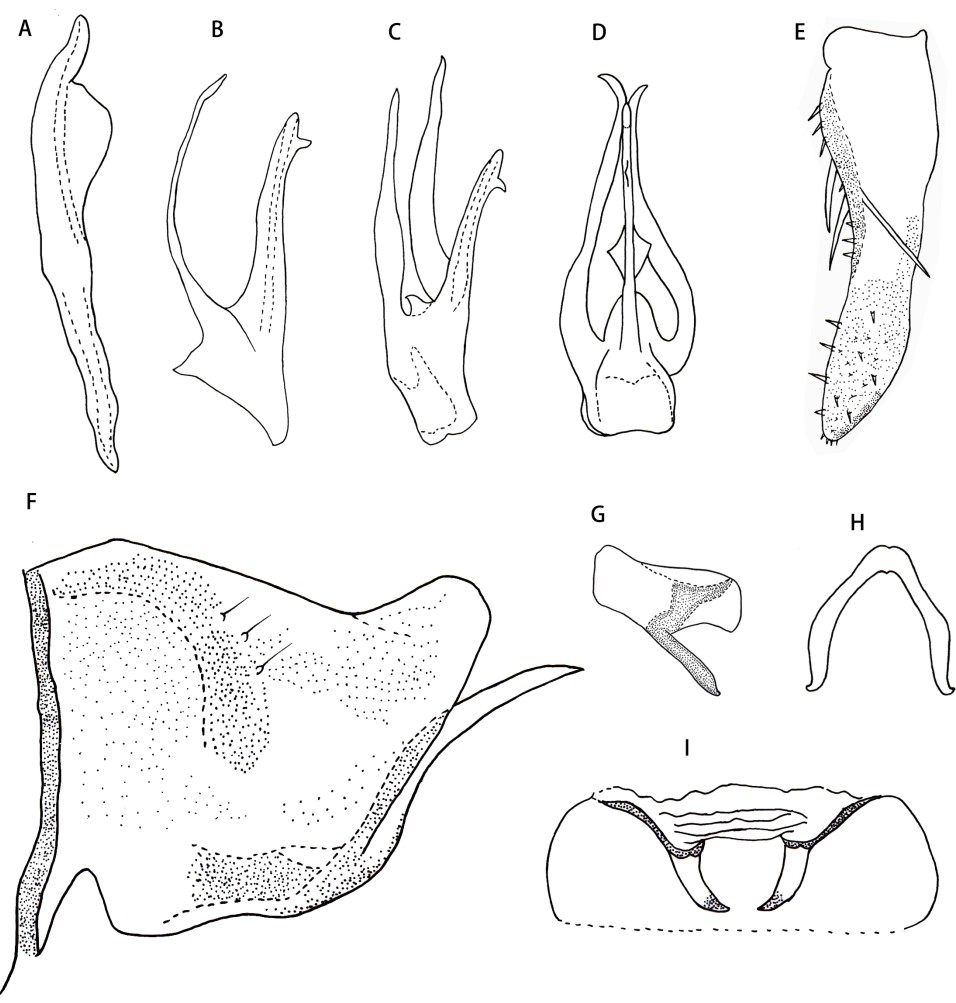

**Figure 4** ***Thaia (Thaia) jiulongensis* Zhang & Song, sp. nov.** (A) Style. (B) Aedeagus, lateral view. (C) Aedeagus, lateral-ventral view. (D) Aedeagus, dorsal view. (E) Subgenital plate. (F) Pygofer lobe, lateral view. (G) Anal tube appendage, lateral view. (H) Connective. (I) Abdominal apodemes.

**Table 6  The location of mismatched base pairs (G-U or U-G) in tRNA from four species.**

|  | *A. (A.) rongchangensis* sp. nov. | *T. (T.) jiulongensis* sp. nov. | *M. bifurcata* | *M. diana* |
|---|---|---|---|---|
| Acceptor arm | trnY, *trnR, trnP* | *trnC, trnG, trnN, trnF,* | *trnY, trnR, trnP* | *trnA, trnC, trnP, trnV, trnY* |
| DHU arm | *trnQ, trnC, trnY, trnG, trnR, trnF, trnH, trnS2, trnL1, trnV* | *trnQ, trnC, trnY, trnG, trnF, trnH, trnP, trnV* | *trnQ, trnC, trnY, trnG, trnR, trnF, trnH, trnS2, trnL1, trnV* | *trnC, trnE, trnF, trnG, trnH, trnP, trnQ, trnS1, trnV* |
| T ψ C arm | *trnW, trnA, trnR, trnS1, trnS2* | *trnR, trnS1, trnT, trnP, trnS2* | *trnW, trnA, trnR, trnS1, trnS2* | *trnA, trnP* |
| Anticodon arm | *trnQ, trnL2, trnH* | *trnC, trnL2, trnH, trnS2* | *trnQ, trnL2, trnH* | *trnH, trnL2* |

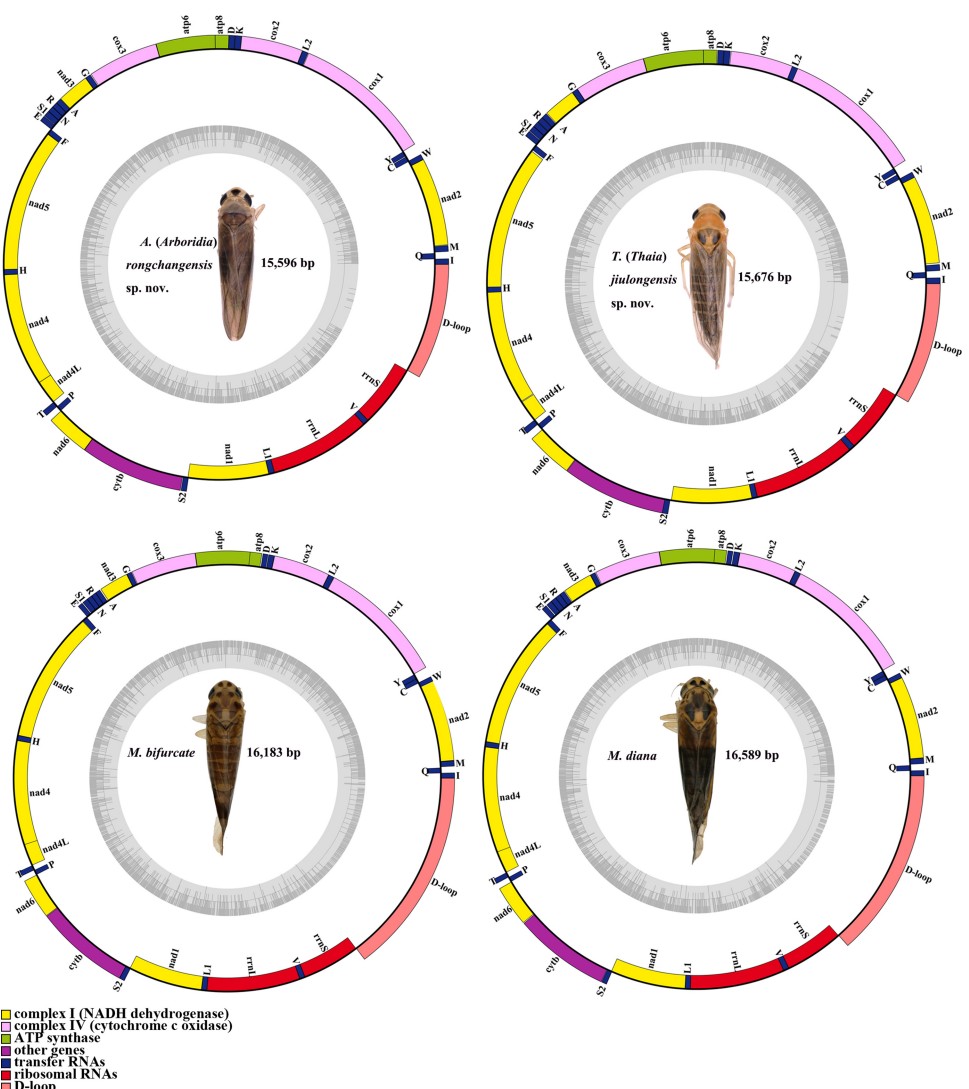

**complex I (NADH dehydrogenase)**
**complex IV (cytochrome c oxidase)**
**ATP synthase**
**other genes**
**transfer RNAs**
**ribosomal RNAs**
**D-loop**

**Figure 5** Mitochondrial map of *A.* (*A.*) *rongchangensis* Zhang & Song, sp. nov., *T.* (*T.*) *jiulongensis* Zhang & Song, sp. nov., *M. bifurcata* and *M. diana*.

## Control region

The control region, also known as the A + T region, acts a crucial part in the size variation of mitogenomes. The largest non-coding regions of the two species, putative control regions, were placed between *rrnS* and *trnI*. The control region in length of *A.* (*A.*) *rongchangensis* sp. nov., *T.* (*T.*) *jiulongensis* sp. nov., *M. bifurcata* and *M. diana* are 1,276 bp, 1,345 bp, 2,472 bp and 2,233 bp, the AT contents are 99.0%, 97.9%, 89.9% and 92.0%, respectively.

## Phylogenetic analysis

In this study, complete mitochondrial genomes from 28 Typhlocybine species were collected as a dataset to establish phylogenetic trees by BI and ML methods, *Bothrogonia ferruginea* and *Iassus dorsalis* were regarded as outgroups. The GenBank accession numbers

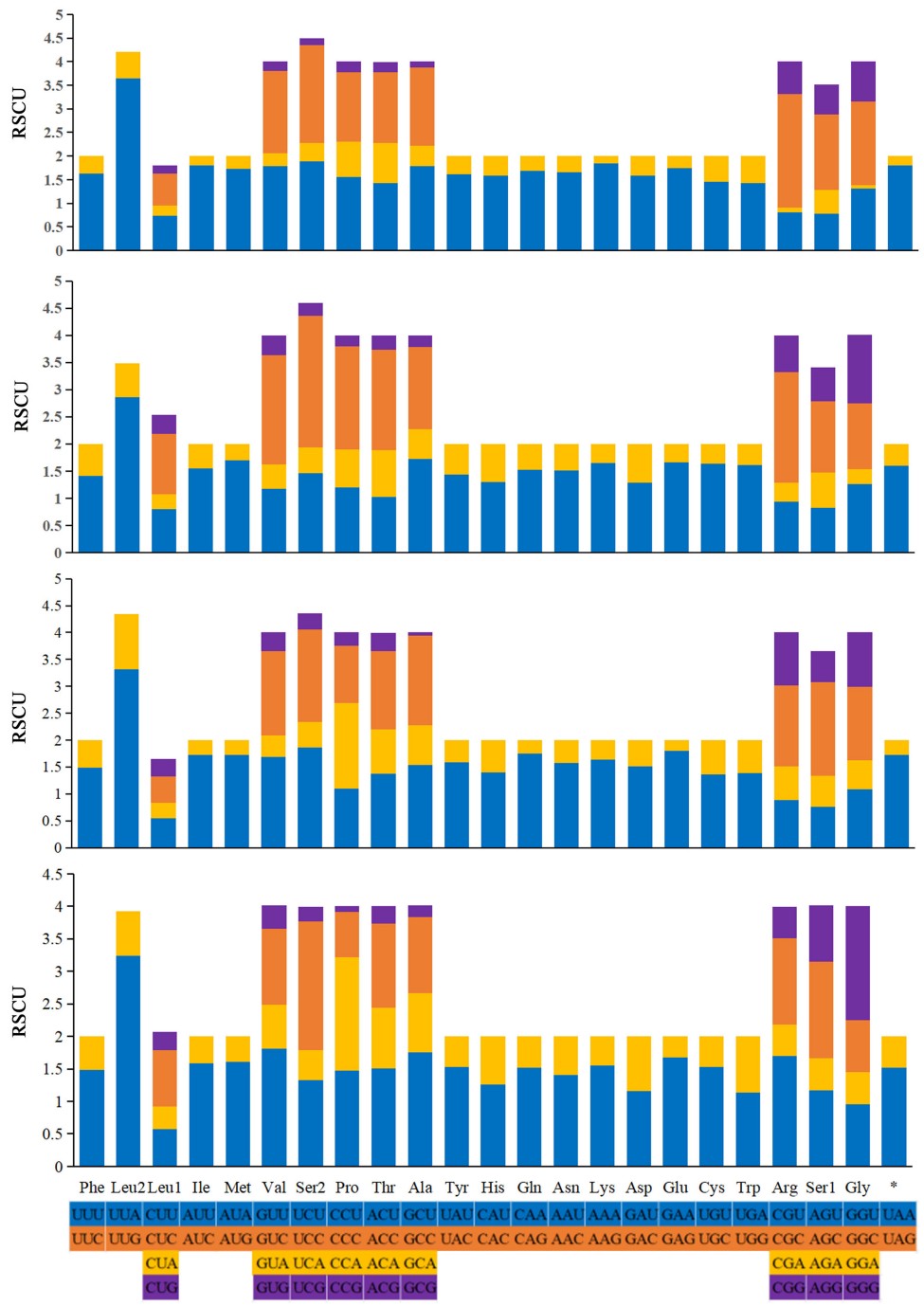

**Figure 6** Relative synonymous codon usage (RSCU) in the mitogenomes of *A.* (*A.*) *rongchangensis* Zhang & Song, sp. nov., *T.* (*T.*) *jiulongensis* Zhang & Song, sp. nov., *M. bifurcata* and *M. diana*.

of all selected species used in this study were listed in Table 4. The phylogenetic topologies constructed by the two methods were completely consistent (Fig. 7). The monophyly of each tribe was generally well supported in the subfamily Typhlocybinae, which is consistent

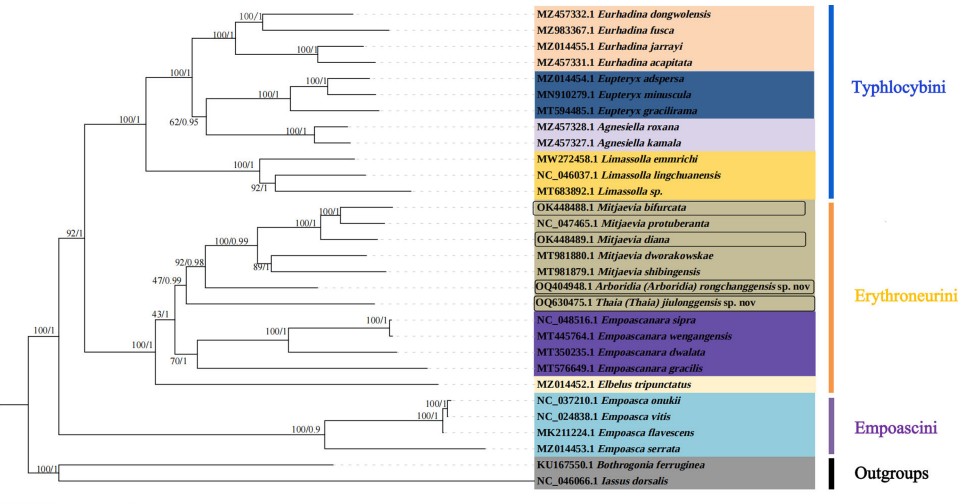

**Figure 7** Phylogenetic tree of Typhlocybinae produced from maximum-likelihood (ML) and Bayesian inference (BI) analyses based on complete mitochondrial gene.

with the findings of some previous molecular phylogenetic studies (*Chen et al., 2021*; *Chen, Li & Song, 2021*). Twelve species of Typhlocybini, twelve species of Erythroneurini, and four species of Empoascini are clustered together, respectively, and all phylogenetic relationships demonstrated higher nodal support in both ML and BI analyses. All species from Typhlocybinae (inner group) are clustered together, all *Mitjaevia* species are gathered together. Our results further confirmed that the genus *Arboridia* has a closer relationship with *Mitjaevia*. Among them, *M. bifurcata*, *M. protuberanta* and *M. diana* are gathered into one clade, while *M. bifurcata* and *M. protuberanta* are sister groups of each other in ML tree and BI tree. In addition, Zyginellini is a junior synonym of Typhlocybini, our result supports recent author's viewpoint (*Dietrich, 2013*; *Zhou, Dietrich & Huang, 2020*; *Yan et al., 2022*). The previous primary diagnosis of Typhlocybinae was made by morphological features, meanwhile, our phylogenetic tree based on molecular data is in agreement with morphological taxonomy. Because the external appearance of *Mitjaevia* species is very similar, the only difference lies in male genitalia including the pygofer, subgenital plate and aedeagus, so, molecular technologies have become particularly important as a supplement to identification of *Mitjaevia* species. This study indicated that mitochondrial genome sequences are the most popularly adopted genomic markers in leafhoppers and becoming increasingly important toward studies in the insect molecular field, involving molecular evolution, phylogeny and phylogeography.

## DISCUSSION

The traditional classification of leafhoppers mainly relies on the morphology of their appearance and male genitalia (*Song & Li, 2013*; *Ramaiah, Meshram & Dey, 2023*; *Xu & Zhang, 2023*). However, due to the large number of leafhoppers and the small size of the Erythroneurini leafhoppers, generally about 2~4 mm (*Dietrich & Dmitriev, 2006*), they

are difficult to identify. In recent years, the development of molecular technology has been applied to the classification of insects (*Singh et al., 2017*; *Lu et al., 2018*; *Matsuia et al., 2022*) and to support the results and correct the attribution of traditional classification. Our findings here on the complete mitochondrial genome supports the classification of two new species.

In addition, we also sequenced and analyzed the mitochondria genomes of two *Mitjaevia* genera, this result enriches the mitochondrial database information of Cicadellidae family and is consistent with the results of previous articles published by our research group (*Chen, Li & Song, 2021*). In the Typhlocybinae, each genus is divided into a separate branch, this result is consistent with previous results (*Lin, Huang & Zhang, 2021*), and all *Mitjaevia* are clustered in one branch. The phylogenetic relationship between the two new species is closer to that of *Mitjaevia*. This may be due to the limited mitochondrial data currently sequenced in Erythroneurini, which requires more and more extensive mitochondrial data to support and elucidate the phylogenetic relationship of the new species. The mitochondrial data can not only confirm the correctness of traditional classification, but also establish a large database, and provide simpler, faster, and more efficient results for subsequent species classification.

## CONCLUSIONS

Two new leafhopper species discovered in Chongqing, *A. (A.) rongchangensis* sp. nov. and *T. (T.) jiulongensis* sp. nov. are described and illustrated. The mitochondrial genomes of these species together with *M. bifurcata* and *M. diana* are assembled and annotated in this work. The study shows that their mitogenomes are conserved in structure, with length of 15,596 bp, 15,676 bp, 16,813 bp and 16,589 bp, including 13 protein-coding genes, 22 tRNA genes, and two rRNA genes. The PCGs begin with ATA/ATG/ATT/TTG, and cease with TAA/TAG/T. All tRNAs are folded into a typical clover-leaf secondary structure, except a few tRNAs with a reduced arm, offering a simple loop or constituted unpaired bases. In this work, the phylogenetic analysis showed a well-supported, *Arboridia* has a closer relationship with *Mitjaevia*. *M. bifurcata*, *M. protuberanta* and *M. diana* are gathered into one clade, while *M. bifurcata* and *M. protuberanta* are sister groups of each other. In addition, Zyginellini can consider as a junior synonym of Typhlocybini. Based on the similarity in appearance of tribe Erythroneurini, the complete mitochondrial genome can provide faster and more convincing evidence for traditional classification.

### Funding

This work was funded by the World Top Discipline Program of Guizhou Province: Karst Ecoenvironment Sciences (No. 125 2019 Qianjiao Keyan Fa), the Innovation Group Project of Education Department of Guizhou Province ([2021]013), the National Natural Science Foundation of China (32260120) and the Natural Science Foundation of Guizhou

Province (Qiankehejichu-ZK [2023] General 257) and the Training Program for High-level Innovative Talents of Guizhou Province (Qiankehepingtairencai-GCC[2023]032). The funders had no role in study design, data collection and analysis, decision to publish, or preparation of the manuscript.

## Grant Disclosures

The following grant information was disclosed by the authors:

World Top Discipline Program of Guizhou Province: Karst Ecoenvironment Sciences: 125 2019 Qianjiao Keyan Fa.

Innovation Group Project of Education Department of Guizhou Province: [2021]013.

National Natural Science Foundation of China: 32260120.

Natural Science Foundation of Guizhou Province: Qiankehejichu-ZK [2023] General 257.

Training Program for High-level Innovative Talents of Guizhou Province: Qiankehepingtairencai-GCC[2023]032.

## Competing Interests

The authors declare there are no competing interests.

## Author Contributions

- Ni Zhang conceived and designed the experiments, performed the experiments, prepared figures and/or tables, authored or reviewed drafts of the article, and approved the final draft.
- Jinqiu Wang conceived and designed the experiments, performed the experiments, authored or reviewed drafts of the article, and approved the final draft.
- Tianyi Pu performed the experiments, analyzed the data, prepared figures and/or tables, and approved the final draft.
- Can Li conceived and designed the experiments, analyzed the data, prepared figures and/or tables, and approved the final draft.
- Yuehua Song conceived and designed the experiments, analyzed the data, authored or reviewed drafts of the article, and approved the final draft.

## DNA Deposition

The following information was supplied regarding the deposition of DNA sequences:

The sequences are available at GenBank: OK448488 (*Mitjaevia bifurcata*), OK448489 (*Mitjaevia diana*), OQ404948 (*Arboridia (Arboridia) rongchangensis* sp. nov) and OQ630475.1 (*Thaia (Thaia) jiulongensis* sp. nov.).

## Data Availability

The sequences are available at NCBI SRA: PRJNA1007413 (*Arboridia (Arboridia) rongchangensis* sp. nov), PRJNA1007448 (*Thaia (Thaia) jiulongensis* sp. nov.).

*Mitjaevia diana* and *Mitjaevia bifurcata* were sequenced by Sanger sequencing.

## New Species Registration

The following information was supplied regarding the registration of a newly described species:

Publication LSID: urn:lsid:zoobank.org:pub:2E7A89DA-21F4-41D1-B9DF-A33BCD5F3086

*Arboridia (Arboridia) rongchangensis* urn:lsid:zoobank.org:act:B290A52A-E70B-44B4-B75D-49E7B62CCA73

*Thaia (Thaia) jiulongensis* urn:lsid:zoobank.org:act:F69EB9C4-388D-4CB8-A045-A8992AE8EB81.

## Supplemental Information

Supplemental information for this article can be found online at http://dx.doi.org/10.7717/peerj.16853#supplemental-information.

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
