# Peer review of "Two new species of Erythroneurini (Hemiptera, Cicadellidae, Typhlocybinae) from southern China based on morphology and complete mitogenomes"

_PeerJ, doi:10.7717/peerj.16853_

## Round 0.1 · original submission · Major Revisions

Please, seriously consider the suggestions of all reviewers, but especially those of Reviewer #3.

Reviewer 1 ·

Basic reporting

Line 37. Number of valid genera and species.
Please see https://hoppers.speciesfile.org/otus/43676/overview
Scroll down to the bottom right for the up-to-date number of valid genera and species in the group based on the World Auchenorrhyncha database.

Line 42 states that leafhoppers could damage plants by spreading of viruses. In fact most of the diseases transmitted by leahoppers are Phytoplasma related diseases, which are bacterial infections.

Line 60. The most comprehensive recent publication on the phylogeny are not cited. See few for example:
Cao, Y.-H., Dietrich, C.H., Kits, J.H., Dmitriev, D.A., Richter, R., Eyres, J., Dettman, J.R., Xu, Y. & Huang, M. (2023) Phylogenomics of microleafhoppers (Hemiptera: Cicadellidae: Typhlocybinae): morphological evolution, divergence times, and biogeography. Insect Systematics and Diversity, 7(4), 1–19. https://doi.org/10.1093/isd/ixad010
Hu, Y., Dietrich, C.H., Skinner, R.K. & Zhang, Y.-L. (2023) Phylogeny of Membracoidea (Hemiptera: Auchenorrhyncha) based on transcriptome data. Systematic Entomology, 48(1), 97–110. [2022] https://doi.org/10.1111/syen.12563
Lu, L., Dietrich, C.H., Cao, Y.-H. & Zhang, Y.-L. (2021) A multi-gene phylogenetic analysis of the leafhopper subfamily Typhlocybinae (Hemiptera: Cicadellidae) challenges the traditional view of the evolution of wing venation. Molecular Phylogenetics and Evolution, 107299. https://doi.org/10.1016/j.ympev.2021.107299
Yan, B., Dietrich, C.H., Yu, X.-F., Jiao, M., Dai, R.-H. & Yang, M.-F. (2022) Mitogenomic phylogeny of Typhlocybinae (Hemiptera: Cicadellidae) reveals homoplasy in tribal diagnostic morphological traits. Ecology and Evolution, 12(e8982), 1–17. https://doi.org/10.1002/ece3.8982
Skinner, R.K., Dietrich, C.H., Walden, K.O., Gordon, E.R.L., Sweet, A.D., Podsiadlowski, L., Petersen, M., Simon, C., Takiya, D.M. & Johnson, K.P. (2019) Phylogenomics of Auchenorrhyncha (Insecta: Hemiptera) using transcriptomes: examining controversial relationships via degeneracy coding and interrogation of gene conflict. Systematic Entomology, 45(1), 85–113. https://doi.org/10.1111/syen.12381

Experimental design

Experimental design is good, the methodology is well described, but the research questions are not well defined.

Validity of the findings

There are several partially linked results presented in the paper.
1. Mitoochondrial gene sequences of 4 species of Erythroneurini are described.
2. The phylogenetic analysis of 24 species based on mitogenome data.
3. Two new species of leafhoppers are described based on morphological data.

Although, each of those 3 aspects of the paper describe important findings, there seems to be no direct linkage between 3 aspects of the paper. Each could potentially be published as independent research project:
2 new species are described exclusively based on morphological data, the species seems to be good species and definitely deserve description.
Study of mitogenomes is a new field of the leafhoper systematics, but there is no any link between this leafhopper morphology.
Phylogenetic tree seem rather artificial addition to the publication. The phylogenetic analysis should be used as a statistical method to test any specific hypotheses. No new hypotheses analyzed in the paper. The phylogenetic tree itself could not be the final result of the research.
After reading the paper, I have feeling that it covers the results of preliminary research.

·

Basic reporting

The text is very verbose. I have tried to shorten it but it probably needs another check. See also Comment 4 below.

Experimental design

The molecular text needs to be reviewed by someone with molecular knowledge, I recommend Chris Dietrich or Jamie Zahniser

Validity of the findings

The morphology is ok with some changes as indicated in my revised Word file. See above and below comments for molecular findings.

Additional comments

A better case for inclusion of the molecular study (which is a large part of the text) needs to be put. The assumption that it is needed to aid classification at the species level is erroneous. At present, molecular studies are more useful for higher classification. It might be better to make the ms just two new species with a little comment on the molecular study. This is one reason I have changed the title.

Regarding the figures,
Are Figs 2b and 4b and c correctly orientated? The text describes the basal process(s) as ventral but in the figures they are dorsal.
Can the subgenital plate figures be put the other way up as is more usual and consistent with other figures.

Reviewer 3 ·

Basic reporting

The core of this paper is the description and illustration of two new leafhopper species from southern China, sequencing and analysis of the mitogenomes of these two species and of two other previously described leafhoppers belonging to the same tribe. These parts are of value although there is nothing particularly remarkable about the new species (both of which belong to diverse genera with many previously described species) or the new mitogenomes (which have the same basic structure and composition of those of most other leafhoppers). The phylogenetic analysis is of little value because, although it shows the position of the newly sequenced species relative to some other members of the same subfamily, the overall taxon sample in the phylogenetic dataset is too limited to provide any important new insights into leafhopper (or even erythroneurine leafhopper) evolution. The literature cited omits two of the three most important and extensive recent studies of typhlocybine phylogeny (Lu et al. 2021, Cao et al 2022). The English also needs extensive editing; the meaning of some sentences is unclear.

Experimental design

There is no clear statement of the research question. The introduction includes statements, e.g., related to the karst habitats of southern China, that seem peripheral to the work performed. The species descriptions and illustrations are good and the descriptions of the mitogenomes follow the standard practices of similar recent publications.

Validity of the findings

The basic findings, i.e., the new species and new mitogenome sequences seem fine. Some other statements, e.g., in the abstract and introduction regarding the major economic importance of these leafhoppers, are not supported by the references cited or by other data. Some species in some groups of leafhoppers are major pests but Erythroneurini includes only a few species that are occasional pests of certain horticultural crops. The Conclusions section is valid but only serves to highlight the limited potential impact of this paper.

---

## Round 0.2 · Minor Revisions

Please, provide minor revisions suggested by the Reviewer 1.

Reviewer 1 ·

Basic reporting

no comment

Experimental design

no comment

Validity of the findings

no comment

Additional comments

Line 33. Please replace the hyperlink for the website citation with proper full citation of the website. The citation could be found at the bottom of the webpage:
https://hoppers.speciesfile.org/otus/43676/overview
Dmitriev, D.A., Anufriev, G.A., Bartlett, C.R., Blanco-Rodríguez, E., Borodin, Oleg I., Cao, Y.-H., Deitz, L.L., Dietrich, C.H., Dmitrieva, M.O., El-Sonbati, S.A., Evangelista de Souza, O., Gjonov, I.V., Gonçalves, A.C., Hendrix, S., McKamey, S., Kohler, M., Kunz, G., Malenovský, I., Morris, B.O., Novoselova, M., Pinedo-Escatel, J.A., Rakitov, R.A., Rothschild, M.J., Sanborn, A.F., Takiya, D.M., Wallace, M.S., Zahniser, J.N. (2022 onward). Erythroneurini Young, 1952. World Auchenorrhyncha Database. TaxonPages. Retrieved on 2023-11-29 at https://hoppers.speciesfile.org/otus/43676/overview
The same change on the line 58.
Line 249. Change ‘anatomy’ to ‘morphology’

---

## Round 0.3 · accepted · Accept

This version of the manuscript can be accepted.